# PGM-Explainer: Probabilistic Graphical Model Explanations for Graph Neural Networks

**Minh N. Vu**
University of Florida
Gainesville, FL 32611
minhvu@ufl.edu

**My T. Thai**
University of Florida
Gainesville, FL 32611
mythai@cise.ufl.edu

## Abstract

In Graph Neural Networks (GNNs), the graph structure is incorporated into the learning of node representations. This complex structure makes explaining GNNs' predictions become much more challenging. In this paper, we propose PGM-Explainer, a Probabilistic Graphical Model (PGM) model-agnostic explainer for GNNs. Given a prediction to be explained, PGM-Explainer identifies crucial graph components and generates an explanation in form of a PGM approximating that prediction. Different from existing explainers for GNNs where the explanations are drawn from a set of linear functions of explained features, PGM-Explainer is able to demonstrate the dependencies of explained features in form of conditional probabilities. Our theoretical analysis shows that the PGM generated by PGM-Explainer includes the Markov-blanket of the target prediction, i.e. including all its statistical information. We also show that the explanation returned by PGM-Explainer contains the same set of independence statements in the perfect map. Our experiments on both synthetic and real-world datasets show that PGM-Explainer achieves better performance than existing explainers in many benchmark tasks.

## 1 Introduction

Graph Neural Networks (GNNs) have been emerging as powerful solutions to many real-world applications in various domains where the datasets are in form of graphs such as social networks, citation networks, knowledge graphs, and biological networks [1, 2, 3]. Many GNN's architectures with high predictive performance should be mentioned are ChebNets [4], Graph Convolutional Networks [5], GraphSage [6], Graph Attention Networks [7], among others [8, 9, 10, 11, 12, 13].

As the field grows, understanding why GNNs made such decisions becomes more vital. (a) It improves the model's transparency and consequently increases trust in the model. (b) Knowledge on model's behaviors helps us identify scenarios in which the systems may fail. This is essential for safety reason in complex real-world tasks in which not all possible scenarios are testable. (c) Due to fairness and privacy reasons, knowing if a model has bias in its decision is crucial. Although there are protection for specific classes of discrimination, there might be other unwanted biases [14]. Understanding the model's decisions helps us discover these biases before its deployment.

Although generating explanations for Conventional Neural Networks (CNNs) has been addressed by many methods and standardized tool-kits, called *explainers* [15, 16, 17, 18, 19, 20, 21, 22, 23], the counterparts for GNNs are lacking. Until very recently, GNNExplainer [24], has been introduced to explain GNNs using a mutual-information approach. As this is a pioneer in explaining GNNs, there is little knowledge on the quality of GNNExplainer and it is unclear whether mutual-information is apt for the task. Furthermore, GNNExplainer requires an explained model to evaluate its prediction on fractional adjacency matrix. However, most available libraries for GNNs, such as Pytorch [25] and DGL [26], do not meet this requirement as the matrix is used to compute the discrete sum in the

messages passing steps. Another work in [27] adapts several existing gradient-based explanation methods for CNNs [15, 18, 22] to GNNs settings. Nevertheless, these methods not only require knowledge on the internal parameters of the model but also are not specifically designed for GNNs. Additionally, all of the above methods fall into a class of explanation methods, named *additive feature attribution methods* [17], which is based on the linearly independent assumption of explained features.[1] However, due to non-linear activation layers, GNN integrates input features in non-linear manner. Relying on linearly independent assumption to generate explanations for GNNs, where explained features can be highly dependent on each other, might degrade the explanation's quality significantly.

**Contribution.** We propose a Probabilistic Graphical Model model-agnostic explainer for GNNs, called PGM-Explainer. In PGM-Explainer, the explanations of a GNN's prediction is a simpler interpretable Bayesian network approximating that prediction. Since Bayesian networks do not rely on the linear-independence assumption of explained features, PGM-Explainer is able to illustrate the dependency among explained features and provide deeper explanations for GNNs' predictions than those of additive feature attribution methods. Our theoretical analysis show that, if a perfect map for the sampled data generated by perturbing the GNN's input exists, a Bayesian network generated by PGM-Explainer always includes the Markov-blanket of the target prediction. This means the resulted PGM contains all statistical information on the explained prediction encoded in the perfect map. We evaluated PGM-Explainer on synthetic datasets and real-world datasets for both node and graph classification. Our evaluation of explainers based on ground-truth explanations and human-subjective tests demonstrate PGM-Explainer provides accurate and intuitive explanations for the predictions.

**Organization.** Section 2 provides some preliminaries, including an explanation model framework [17] and our discussion on the selection of PGM as interpretable models explaining GNNs. Detailed description of PGM-Explainer is provided in Section 3. Our experimental evaluation on performance of different explainers is reported in Section 4. Finally, Section 5 concludes our paper.

## 2 Preliminaries

Given a GNN model $\Phi$ and a target to be explained $t$, let $\Phi_t : \mathcal{G} \to \mathcal{K}$ be a prediction to be explained. Here, $\mathcal{G}$ is the set of all possible input graphs of the model and $\mathcal{K}$ is the classification's space. In a node classification, $\Phi(G)$ is the vector of predictions on all nodes of $G \in \mathcal{G}$ and $\Phi_t(G) \equiv \Phi(G)_t$ is the target prediction. For a graph classification, $\Phi(G)$ is the prediction on $G$ and we simply set $\Phi_t(G)$ to be $\Phi(G)$. In GNN, each input graph $G = (V, E)$ with $F$ features on each node is feed into $\Phi$ using the $|V| \times F$ features matrix $X$ and the $|V| \times |V|$ adjacency matrix $A$. In this work, we consider the black-box model where explainers do not have any information on the internal of $\Phi$. Specifically, we allow explainers to observe different predictions by performing multiple queries on $\Phi$; however, back-propagation and similar operations based on the model's parameters are not allowed. Regarding our notations, when we associate a graph's component, for example a node $v$, with a random variable, we use bold notation, such as $\boldsymbol{v}$, to emphasize the distinction between them.

**Explanation Model.** An explanation $\zeta$ of $\Phi_t$ is normally drawn from a set of possible explanations, called interpretable domain $\mathcal{E}$. Selecting $\mathcal{E}$ in explaining GNN's prediction can vary from a subset of edges of $G$ [24] to a subset of entries in $X$ [27]. In this work, we adopt an explanation model framework proposed in [17] for neural networks and consider $\mathcal{E}$ to be a family of interpretable models. In this explanation model, given an objective function $R_{\Phi,t} : \mathcal{E} \to \mathbb{R}$ associating each explanation with a score, the explanation can be considered as the solution of the following optimization problem:

$$\zeta^* = \arg\max_{\zeta \in \mathcal{E}} R_{\Phi,t}(\zeta). \tag{1}$$

To encourage a compact solution $\zeta^*$, explainers might introduce some constraints on (1). We use a general condition $\zeta \in \mathcal{C}$ where $\mathcal{C} \subseteq \mathcal{E}$ to represent these constraints. For instance, we can promote simpler model $\zeta^*$ by setting $\mathcal{C}$ to be a set of models with a limited number of free parameters.

**Probabilistic Graphical model as Interpretable Domain.** Selecting an appropriate interpretable domain $\mathcal{E}$ is crucial to the explainer's quality. The first reason is the trade-off in the complexity of $\mathcal{E}$. On one hand, $\mathcal{E}$ must be complex enough to explain the target prediction. On the other hand, $\mathcal{E}$

should be simple so that end-users can interpret the explanation. Intuitively, for each $\zeta \in \mathcal{E}$ to explain $\Phi_t$ faithfully, the behavior of $\zeta$ must be similar to that of $\Phi_t$. Hence, in order to explain GNNs, $\mathcal{E}$ must be chosen such that it contains models having similar behaviors to that of GNNs' predictions.

Probabilistic Graphical models (PGMs) [28] are statistical models encoding complex distributions over a multi-dimensional space using graph-based representation. In general, the probabilistic graph of a distribution represents the distribution compactly in a factorized form. Knowing the PGM not only allows the distribution to be written down tractably but also provides a simple interpretation of the dependencies of those underlying random variables. As our target of explanation is a complex function $\Phi_t$ with high dependencies among input features, approximating $\Phi_t$ by linear functions can lead to poor results. In contrast, PGM encodes rich sets of information on the graph's components which can potentially support us analyzing the contributions of each component toward the examined variable. For instance, PGM is able to tell us certain explained features can determine the target prediction only under specific realizations of some other features. Such kind of information clearly cannot be obtained from linear models.

Bayesian network [29], a PGM representing the conditional dependencies among variables via a directed acyclic graph, is one of the most well-known PGM due to its intuitive representation. Furthermore, efficient algorithms searching for Bayesian network given sampled data, known as structure learning, have been studied extensively [28, 30]. As such, given target of explanation $\Phi_t$, our proposed PGM explanation is the optimal Bayesian network $\mathcal{B}^*$ of the following optimization:

$$\arg \max_{\mathcal{B} \in \mathcal{E}} R_{\Phi,t}(\mathcal{B}), \quad \text{s.t.} \ |\mathcal{V}(\mathcal{B})| \leq M, \boldsymbol{t} \in \mathcal{V}(\mathcal{B}), \tag{2}$$

where $\mathcal{E}$ is the set of all Bayesian networks. $\mathcal{V}(\mathcal{B})$ is the set of random variables in Bayesian network $\mathcal{B}$ and $\boldsymbol{t}$ is the random variable corresponding to the target prediction $t$. In optimization (2), the first constraint ensures that the number of variables in $\mathcal{B}$ is bounded by a given constant $M$ to encourage a compact solution and the second constraint guarantees the target prediction is included in the explanation.

In PGM-Explainer, we also support the searching for PGM with the following *no-child* constraint:

$$\text{Ch}_{\mathcal{B}}(\boldsymbol{t}) = \emptyset \tag{3}$$

where $\text{Ch}_{\mathcal{B}}(\boldsymbol{t})$ is the set of children of node $\boldsymbol{t}$ in Bayesian network $\mathcal{B}$. We introduce this constraint because, in some cases, a target variable $\boldsymbol{t}$ is more desirable to be a leaf in a PGM explanation. This condition not only let us answer conditional probability queries on target variables more efficiently, but also makes the resulted PGM more intuitive. For instance, in a Bayesian network, the parents of the same child can directly influence the distributions of each other even though there might be no edge among them. This additional constraint allows us to avoid this ambiguity in explaining the target variable. We provide illustrative example of this *no-child* constraint in Appendix B.

## 3 PGM-Explainer: Probabilistic Graphical Model Explanations for GNNs

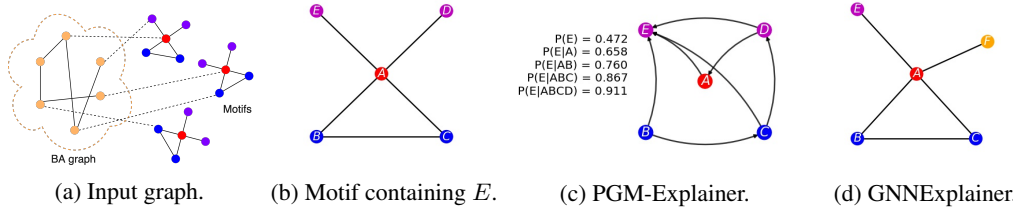

| (a) Input graph. | (b) Motif containing $E$. | (c) PGM-Explainer. | (d) GNNExplainer. |

Figure 1: Example of PGM-explanation of a GNN's prediction. **(a)** An input graph of the GNN model: a combination of a 300-node BA graph, 80 5-node motifs and some random edges. The labels of nodes are assigned based on their roles in the graph (shown in color) **(b)** A motif containing the target prediction to be explained. We aim to explain the prediction of the role of node $E$. **(c)** A PGM-explanation in a form of Bayesian network. This Bayesian network estimates the probability that node $E$ has the predicted role given a realization of other nodes. **(d)** An explanation of GNNExplainer, which mistakenly includes node $F$ in the BA graph.

**Illustrative Example.** We demonstrate an example of PGM explanation in Fig 1. We adopt the test approach of [24] where the input graph is a combination of a Barabási-Albert (BA) graph, a set of

motifs and some random edges.[2] Nodes are assigned into four classes based on their roles as shown by different color in Fig. 1a. A ground-truth explanation of a prediction on a node in a motif is all the nodes in the motif. In this example, the target of explanation is the role "purple" of node $E$ in Fig. 1b. Our PGM-Explainer is able to identify all nodes in the motif and constructs a PGM approximating the target prediction (Fig. 1c). Different from existing explainers, PGM-Explainer provides statistical information on the contributions of graph's components in term of conditional probabilities. For example, without knowing any information on $E$'s neighborhood, the PGM explanation approximates a probability of predicting $E$ to be "purple" is $47.2\%$. If PGM knew a prediction of node $A$ and its realization, that probability is increased to $65.8\%$. This information not only helps users evaluate the contribution of each explained features on the target prediction but also provides intuition on their interactions in constituting that prediction.

**Major Components of PGM-Explainer.** PGM-Explainer consists of three major steps: data generation, variables selection, and structure learning, which are summarized in Fig. 2. The data generation step is to generate, preprocess and record a set of input-output pairs, called sampled data, of the prediction to be explained. The variables selection step eliminates unimportant variables from the sampled data to improve the running time and encourage compact explanations. The final step, structure learning, takes the filtered data from the previous step and generates a PGM explanation.

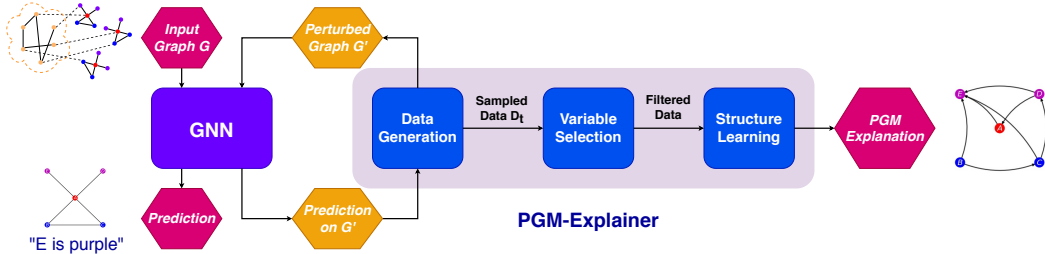

Figure 2: The architecture of PGM-Explainer. Given input graph $G$ and a prediction to be explained, PGM-Explainer generates perturbed graphs and records GNN's predictions on those graphs in the data generation step. The variable selection step eliminates unimportant explained features in this data and forwards the filtered data. Finally, the PGM is generated in the structure learning step.

## 3.1 Data Generation

The goal of the data generation step in PGM-Explainer is to generate a set of sampled data $\mathcal{D}_t$ from the target function $\Phi_t$. In the consequence steps, the PGM will be learnt from this sampled data $\mathcal{D}_t$. Since the explainer aims to capture the behaviors of $\Phi_t$, especially at the neighborhoods of input graph $G$, PGM-Explainer first perturbs some features of $G$ to obtain a set of perturbed samples. Specifically, we fix a parameter $p \in (0, 1)$ representing a probability that the features in each node is perturbed. For each node $v$ in $G$, we introduce a random variable $s_v$ indicating whether the features on $v$ is perturbed. The perturbation scheme might vary depending on applications since we want the perturbed graph to be in the vicinity of $G$. We implement different perturbing algorithms; however, the scheme used in this paper is simply setting the node features to the mean value among all nodes. For each realization of $\mathbf{s} = \{s_v\}_{v \in V}$, we obtain an induced graph $G(\mathbf{s}) \in \mathcal{G}$. The prediction $\Phi(G(\mathbf{s}))$ on the induced graph is then obtained by feeding $G(\mathbf{s})$ through the GNN.

In a node classification task, for each perturbation, the realization of node variable $\boldsymbol{v} = \{s_v, \mathrm{I}(\Phi(G(\mathbf{s}))_v)\}$ is recorded into $\mathcal{D}_t$, where $\mathrm{I}(.)$ is a function indicating whether the prediction $\Phi(G(\mathbf{s}))_v$ is different significantly from the original prediction $\Phi(G)_v$. Intuitively, $\boldsymbol{v}$ encodes both the influence of features of node $v$ onto the graph's prediction and the influence of the overall changes in the graph to the prediction on $v$. In our implementations of PGM-Explainer, $s_v$ and $\mathrm{I}(.)$ are stored in binary values and the domain of $\boldsymbol{v}$ is consequently implemented using only two bits. If the GNN has $L$ graph neural network layers, the prediction on $t$ can only be influenced by $L$-hop neighbors of $t$. Hence, the explainer only needs to examine the $L$-hop neighbors of $t$. Thus, for each realization $\mathbf{s}$, PGM-Explainer only records $\boldsymbol{v}$ for $v \in \mathrm{Ne}_t^G$ where $\mathrm{Ne}_t^G$ is $L$-hop neighbors of $t$ in $G$. After $n$

samplings of **s**, PGM-Explainer will obtain a data table $\mathcal{D}_t$ of size $n \times |\text{Ne}_t^G|$, where each entry $\mathcal{D}_{t_{iv}} = \boldsymbol{v}^{(i)}$ is the $i^{th}$ sampling of node random variable $\boldsymbol{v}$.

For a graph classification task, we set the node variable $\boldsymbol{v} = \{s_v\}$ and introduce an additional target variable $\boldsymbol{t} = \{\text{I}(\Phi(G(\mathbf{s})))\}$ into the set of random variables. Furthermore, we set the neighborhood set $\text{Ne}_t^G$ to be $V$ since all nodes on the input graph can influence the graph prediction. Therefore, the size of $\mathcal{D}_t$ in this case is $n \times (|V| + 1)$.

### 3.2 Variables Selections

The task of learning PGM $\mathcal{B}$ from $\mathcal{D}_t$ is called structure learning. Unfortunately, finding an optimal PGM from sampled data is intractable [28]. For a typical GNN, set $\text{Ne}_t^G$ might contain thousands of nodes and searching for an optimal Bayesian network is very expensive. This challenge requires us to further trim the set of variables to be examined by the structure learning algorithms.

To reduce the number of variables in $\mathcal{D}_t$, we need to identify which variables are important and avoid eliminating them. PGM-Explainer addresses this problem by observing that important variables to target variable $\boldsymbol{t}$ are in the Markov-blanket of $\boldsymbol{t}$. By definition, the Markov-blanket of $\boldsymbol{t}$ in a PGM $\mathcal{P}$, denoted as $\text{MB}_{\mathcal{P}}(\boldsymbol{t})$, is the minimum set of variables such that, given realizations of all variables in the Markov-blanket, $\boldsymbol{t}$ is conditionally independent from all other variables in $\mathcal{P}$.[3] Therefore, under an assumption that there exists a perfect map $\mathcal{B}^*$ for the distribution of random variables in $\text{Ne}_t^G$, analyzing $\text{MB}_{\mathcal{B}^*}(\boldsymbol{t})$ provides us the same statistical information on $\boldsymbol{t}$ as in $\text{Ne}_t^G$.[4] Thus, instead of finding $\mathcal{B}^*$ on $\text{Ne}_t^G$, we can determine $\text{MB}_{\mathcal{B}^*}(\boldsymbol{t})$ and compute PGM $\mathcal{B}$ on $\text{MB}_{\mathcal{B}^*}(\boldsymbol{t})$ as an explanation for $\Phi_t$. Due to the property of Markov-blanket, statistical information of $\boldsymbol{t}$ in $\mathcal{B}$ and $\mathcal{B}*$ are the same.

Markov-blanket $\text{MB}_{\mathcal{B}^*}(\boldsymbol{t})$ can be obtained by the Grow-Shrink (GS) algorithm [32]. However, the number of conditional independent tests in GS can be in an exponential order of the number of random variables. Fortunately, in order to generate explanations for GNN's predictions, the explainer does not need to know exactly $\text{MB}_{\mathcal{B}^*}(\boldsymbol{t})$. In fact, knowing any set containing $\text{MB}_{\mathcal{B}^*}(\boldsymbol{t})$ is sufficient for the resulted PGM to contain all the information of the target prediction. In PGM-Explainer, we propose to find a small subset $\boldsymbol{U}(\boldsymbol{t})$ of $\text{Ne}_t^G$ which is guaranteed to contain $\text{MB}_{\mathcal{B}^*}(\boldsymbol{t})$. A definition of such $\boldsymbol{U}(\boldsymbol{t})$ and the guarantee that $\text{MB}_{\mathcal{B}^*}(\boldsymbol{t}) \subseteq \boldsymbol{U}(\boldsymbol{t})$ is given in Theorem 1 (proof in Appendix D):

**Theorem 1** *Assume there exists a perfect map $\mathcal{B}^*$ of a distribution $P$ on a set of random variables $\boldsymbol{V}$. For any random variable $\boldsymbol{v}$ in $\boldsymbol{V}$, we denote $\boldsymbol{S}(\boldsymbol{v}) = \{\boldsymbol{v}' \in \boldsymbol{V} | \boldsymbol{v}' \not\perp \boldsymbol{v}\}$. Then, for any $\boldsymbol{t} \in \boldsymbol{V}$, we have $\text{MB}_{\mathcal{B}^*}(\boldsymbol{t}) \subseteq \boldsymbol{U}(\boldsymbol{t})$ where $\boldsymbol{U}(\boldsymbol{t}) = \cup_{\boldsymbol{v} \in \boldsymbol{S}(\boldsymbol{t})} \boldsymbol{S}(\boldsymbol{v})$.*

Based on Theorem 1, constructing $\boldsymbol{U}(\boldsymbol{t})$ from $\mathcal{D}_t$ is straightforward. PGM-Explainer first computes $O\left(|\text{Ne}_t^G|^2\right)$ independent tests to construct $\boldsymbol{S}(\boldsymbol{v}) \,\forall, \boldsymbol{v}$. Then $\boldsymbol{U}(\boldsymbol{t})$ is obtained by combining all elements of $\boldsymbol{S}(\boldsymbol{v}), \forall \boldsymbol{v} \in \boldsymbol{S}(\boldsymbol{t})$. In case *no-child* constraint (3) is considered, we can solve for a much smaller set $\boldsymbol{U}(\boldsymbol{t})$ containing $\text{MB}_{\mathcal{B}^*}(\boldsymbol{t})$ based on Theorem 2 (proof in Appendix E):

**Theorem 2** *Assume there exists a perfect map $\mathcal{B}^*$ of a distribution $P$ on $\boldsymbol{V}$. If node $\boldsymbol{t} \in \boldsymbol{V}$ has no child in $\mathcal{B}^*$, $\text{MB}_{\mathcal{B}^*}(\boldsymbol{t}) \subseteq \boldsymbol{U}(\boldsymbol{t})$ where $\boldsymbol{U}(\boldsymbol{t}) = \boldsymbol{S}(\boldsymbol{t}) \triangleq \{\boldsymbol{v} | \boldsymbol{v} \not\perp \boldsymbol{t}\}$.*

Note that, in this case, we can learn the set $\boldsymbol{U}(\boldsymbol{t})$ using only $O\left(|\text{Ne}_t^G|\right)$ independent tests.

Given $\boldsymbol{U}(\boldsymbol{t})$, the explainer can learn the PGM on top of $\boldsymbol{U}(\boldsymbol{t})$ instead of on $|\text{Ne}_t^G|$ random variables. We want to emphasize that the $\boldsymbol{U}(\boldsymbol{t})$'s construction in both Theorem 1 and 2 of PGM-Explainer only use pairwise dependence tests, not conditional dependence tests as in conventional algorithms solving for the Markov-blankets. This is a significant difference since a conditional dependence test conditioning on $m$ binary variables further requires $2^m$ dependence tests. Two variables are declared dependent if any of these dependency tests asserts that the distributions are different. When the number of data samples is limited, the probability of including wrong variables into the Markov-blanket is rapidly increasing with the conditioning set size [32]. Additionally, by using the dependency tests only, we can compute $\boldsymbol{U}(\boldsymbol{v})$ for all $\boldsymbol{v}$ in the PGM. Thus, for node classification tasks, PGM-Explainer is able to generate batch explanations for many target predictions simultaneously.

### 3.3 Structure Learning

The final step of PGM-Explainer is to learn explanation Bayesian network $\mathcal{B}$ from $\mathcal{D}_t$. We first demonstrate the learning without constraint (3). We propose to use the *BIC score* as follows:

$$R_{\Phi,t}(\mathcal{B}) = \text{score}_{BIC}(\mathcal{B} : \mathcal{D}_t[\boldsymbol{U}(\boldsymbol{t})]) = l(\hat{\theta}_{\mathcal{B}} : \mathcal{D}_t[\boldsymbol{U}(\boldsymbol{t})]) - \frac{\log n}{2}\text{Dim}[\mathcal{B}] \qquad (4)$$

where $\mathcal{D}_t[\boldsymbol{U}(\boldsymbol{t})]$ is data $\mathcal{D}_t$ on variables $\boldsymbol{U}(\boldsymbol{t})$ and $\text{Dim}[\mathcal{B}]$ is the dimension of model $\mathcal{B}$. $\theta_{\mathcal{B}}$ are the parameters of $\mathcal{B}$ and function $l(\theta_{\mathcal{B}} : \mathcal{D}_t[\boldsymbol{U}(\boldsymbol{t})])$ is the log-likelihood between $\mathcal{D}_t[\boldsymbol{U}(\boldsymbol{t})]$ and $\theta_{\mathcal{B}}$, i.e. $l(\theta_{\mathcal{B}} : \mathcal{D}_t[\boldsymbol{U}(\boldsymbol{t})]) = P(\mathcal{D}_t[\boldsymbol{U}(\boldsymbol{t})]|\theta_{\mathcal{B}}, \mathcal{B})$. $\hat{\theta}_{\mathcal{B}}$ in (4) is parameters' value that maximizes the log-likelihood, which is called the maximum likelihood estimator. Given this objective, PGM-Explainer can use exhaustive-search to solve for an optimal PGM. However, we observe that hill-climbing algorithm [33] also returns good local optimal solutions with a significantly lower running time.

In PGM-Explainer, we chose the *BIC score* objective because this objective is proven to be *consistent* with the data [28]. A scoring function is consistent if, as the number of samples $n \to \infty$, the two followings conditions hold: (i) $\mathcal{B}^*$ maximizes the score and (ii) all structures that do not contain the same set of independencies with $\mathcal{B}^*$ have strictly lower scores. The second condition is also known as the *I-equivalent* condition. These two properties imply that *BIC score* asymptotically prefers a structure that exactly fits the dependencies in the data. Consequently, with large enough samplings, Bayesian network $\mathcal{B}$ obtained by maximizing the *BIC score* should reflect the dependencies of variables in $\boldsymbol{U}(\boldsymbol{t})$ and thus provides us the reasons behind the model's decision. In Appendix C.3, we provide further intuition and more formal discussion on the *BIC score*.

The pseudo-code of PGM-Explainer without no-child constraint is shown in Alg. 1, Appendix F. In summary, first, data $\mathcal{D}_t$ is generated by random perturbations on input features. Then, PGM-Explainer trims down the number of variables using pairwise independence tests. Finally, the PGM is learnt using *BIC score* with hill-climbing algorithm.

To impose the *no-child* constraint (3), we need to modify the structure learning step to ensure that solutions remain consistent with the data, as shown in Alg. 2, Appendix G. Instead of maximizing *BIC score* on $\mathcal{D}_t[\boldsymbol{U}(\boldsymbol{t})]$, PGM-Explainer computes an optimal PGM $\mathcal{B}'$ on $\mathcal{D}'_t = \mathcal{D}_t[\boldsymbol{U}(\boldsymbol{t}) \setminus \{\boldsymbol{t}\}]$. Then, the set of $\boldsymbol{t}$'s parents is obtained by iteratively removing variables from $\boldsymbol{U}(\boldsymbol{t}) \setminus \{\boldsymbol{t}\}$. This can be done in a similar manner as the shrinking step in GS [32]. After that, PGM-Explainer includes $\boldsymbol{t}$ back into $\mathcal{B}'$ and adds directed edges from all parents to $\boldsymbol{t}$ to get the final explanation. The following Theorem shows that, the obtained PGM $\hat{\mathcal{B}}$ is *I-equivalence* to $\mathcal{B}^*$.

**Theorem 3** *Assume there exists a perfect map $\mathcal{B}^*$ of a distribution $P$. For a variable $\boldsymbol{t}$ having no child in $\mathcal{B}^*$, $\mathcal{D}'_t$ is the set of $n$ sampling data from $P$ without the data for $\boldsymbol{t}$, $\text{Pa}_{\mathcal{B}^*}(\boldsymbol{t})$ is the set of parents of $\boldsymbol{t}$ in $\mathcal{B}^*$ and $\hat{\mathcal{B}}'$ is an optimal solution of $\max_{\mathcal{B}'} \text{score}_{BIC}(\mathcal{B}' : \mathcal{D}'_t)$. As $n \to \infty$, a Bayesian network $\hat{\mathcal{B}}$ obtaining by adding node $\boldsymbol{t}$ to $\hat{\mathcal{B}}'$ and adding edges $(\boldsymbol{v}, \boldsymbol{t})$ for all $\boldsymbol{v} \in \text{Pa}_{\mathcal{B}^*}(\boldsymbol{t})$ is I-equivalence to $\mathcal{B}^*$ with probability 1.*

Proof of Theorem 3 is presented in Appendix H. Note that if we run Alg. 2 on a target $\boldsymbol{t}$ that has children in the optimal PGM $\mathcal{B}^*$, we will obtain a PGM $\mathcal{B}$ that contains no independence statement that is not in $\mathcal{B}^*$. Furthermore, the Markov-blanket of $\boldsymbol{t}$ in $\mathcal{B}^*$ is the set of parents of $\boldsymbol{t}$ in $\mathcal{B}$. With this, we have finalized the description of PGM-Explainer with *no-child* constraint.

## 4 Experiments

This section provides our experiments, comparing the performance of PGM-Explainer to that of existing explanation methods for GNNs, including GNNExplainer [24] and our implementation of the extension of SHapley Additive exPlanations (SHAP) [17] to GNNs. We select SHAP to represent the gradient-based methods because of two reasons. First, source codes of gradient-based methods for GNNs are either unavailable or limited to specific models/applications. Second, SHAP is an *additive feature attribution methods*, unifying explanation methods for conventional neural networks [17]. By comparing PGM-Explainer with SHAP, we aim to demonstrate drawbacks of the linear-independence assumption of explained features in explaining GNN's predictions. We also show that the vanilla gradient-based explanation method and GNNExplainer can be considered as *additive feature attribution methods* in Appendix A. Our source code can be found at [34].

### 4.1 Datasets and Experiment Settings

**Synthetic node classification task.** Six synthetic datasets, detailed in Appendix I, were considered. We reuse the source code of [24] as we want to evaluate explainers on the same settings. In these datasets, each input graph is a combination of a base graph and a set of motifs. The ground-truth label of each node on a motif is determined based on its role in the motif. As the labels are determined based on the motif's structure, the explanation for the role's prediction of a node are the nodes in the same motif. Thus, the ground-truth explanation in these datasets are the nodes in the same motif as the target. Since the ground-truth explanations are clear, we use accuracy as an evaluation metric.

**Real-world node classification task.** We use the *Trust weighted signed networks* Bitcoin-Alpha and Bitcoin-OTC datasets [35]. These are networks of 3783 and 5881 accounts trading Bitcoin on platforms called Bitcoin-Alpha and Bitcoin-OTC respectively. In each network, members rate other members in a scale of -10 (total distrust) to +10 (total trust). We label each account *trustworthy* or *not-trustworthy* based on those ratings. For each account, its features encode the account's outgoing information such as the average rate made by that account or the normalized number of votes that account made. Since each account is rated by many others, we do not know exactly which neighbors' information are exploited by the GNN to make its predictions, we evaluate explanations based on precision instead of accuracy. Any accounts in an explanation that do not rate the target or rate the target negatively are counted as "wrong accounts". The precision of an explanation is computed by dividing the "not wrong accounts" by the total number of accounts in that explanation.

**Real-world graph classification task.** We use MNIST SuperPixel-Graph dataset [36] of which each sample is a graph corresponding to an image sample in the hand-written digit MNIST dataset [37]. In this dataset, the original MNIST images are converted to graphs using super-pixels, which represent small regions of homogeneous intensity in images. In a given graph, each node represents a super-pixel in the original image while the node's features are the intensity and locations of the super-pixel (see Fig. 5(a) for examples). As the ground-truth explanations for this experiment are not available, we use human-subjective test to evaluate the results of explainers.

**Model and experiments setup.** For each dataset, we use Adam optimizer [38] to train a single GNN for the corresponding task. For the synthetic datasets, the Bitcoin-Alpha dataset and Bitcoin-OTC dataset, we use the GNN models provided by [24]. All models have 3 graph layers with the number of parameters between $1102$ and $1548$. The train/validation/test split is $80/10/10\%$. The models are trained for 1000 epochs with learning rate 0.001. For the MNIST SuperPixel-Graph dataset, the dataset is divided into 55K/5K/10K. We use the Graph Convolutional Network (GCN) model [5] implemented by [36] due to its popularity and efficiency in training. The model has 101365 parameters and converges at epoch $188$ with learning rate is automatically chosen between $10^{-3}$ and $10^{-4}$. The models' test-set accuracy and the number of samples $n$ used by PGM-Explainer to generate the explanations are reported in Table 1.

Table 1: Models' accuracy and number of sampled data used by PGM-Explainer

| | Node Classification | | | | | | | | Graph Classification |
|---|---|---|---|---|---|---|---|---|---|
| *Experiment* | **Syn 1** | **Syn 2** | **Syn 3** | **Syn 4** | **Syn 5** | **Syn 6** | **Bitcoin-alpha** | **Bitcoin-OTC** | **GCN-MNIST** |
| *Accuracy* | 97.9 | 85.4 | 100.0 | 99.4 | 89.1 | 99.3 | 93.9 | 89.5 | 90.4 |
| *No. Samples* | 800 | 800 | 800 | 1600 | 4000 | 800 | 1000 | 1000 | 400 |

### 4.2 Results on Synthetic Datasets

Table 2 shows the accuracy in the explanations generated by different explainers. Here the explanations are generated for all nodes in the motifs of the input graph. All explainers are programmed to return the top $k$ important nodes to the target prediction where $k$ is the number of nodes in the target's motif. For GNNExplainer, the accuracy we obtained using its default implementation provided in [39] is lower than that reported in the corresponding paper [24]. Perhaps the tuning parameters of GNNExplainer used in [24] are not available to us via [39]. To be consistent, we report the original results in black and our results in blue for GNNExplainer.

As can be seen, PGM-Explainer outperforms other explanation methods, especially on dataset 2 and 6. We specifically design the dataset 6 so that the role of a node on a motif cannot be determined by knowing only one of its neighbors. Furthermore, for some nodes, such as node $D$ and $E$ in Fig. 1b, the roles can only be determined using almost all 2-hop neighbors. To achieve high predictive

Table 2: Accuracy of Explainers on Synthetic Datasets.

| Explainer | Syn 1 | Syn 2 | Syn 3 | Syn 4 | Syn 5 | Syn 6 |
|---|---|---|---|---|---|---|
| SHAP | 0.947 | 0.741 | 0.872 | 0.884 | 0.641 | 0.741 |
| GNNExplainer | 0.925 - 0.729 | 0.836 - 0.750 | 0.741 | 0.948 - 0.862 | 0.875 - 0.842 | 0.612 |
| PGM-Explainer | 0.965 | 0.926 | 0.885 (0.942) | 0.954 (0.968) | 0.878 (0.892) | 0.953 |

performance on this dataset, the GNN must integrate network features in a non-linear manner. Specifically, observing only red node $A$ does not give the GNN much information on the target. Knowing one of two blue nodes $B$ and $C$ do not provide much information for the prediction either. However, the knowledge on all those three nodes can help the GNN fully determine the role of $E$. To faithfully explain the GNN, an explainer must be able to capture these non-linear behaviors.

For some synthetic datasets, we observe that GNNs can determine the target node's role without knowing all nodes on the target's motif. Thus allowing explainers to return less than $k$ nodes can improve the overall precision. PGM-Explainer can easily make this decision by performing dependence tests on those $k$ nodes. We report the precision in parentheses for PGM-Explainer.

### 4.3 Results on Real-world Datasets

The precision of nodes in explanations of each explainer on Trust weighted signed networks datasets are reported in Table 3. Here we test the total number of accounts in each explanation at 3, 4 and 5. Our results show that PGM-Explainer achieves significantly higher precision than other methods in these experiments.

Table 3: Precision of Explainers on Trust Signed networks datasets

| Explainer | Bitcoin-alpha | | | Bitcoin-OTC | | |
|---|---|---|---|---|---|---|
| | top 3 | top 4 | top 5 | top 3 | top 4 | top 5 |
| SHAP | 0.537 | 0.498 | 0.465 | 0.607 | 0.587 | 0.566 |
| GNNExplainer | 0.375 | 0.332 | 0.307 | 0.385 | 0.338 | 0.312 |
| PGM-Explainer | 0.873 | 0.857 | 0.848 | 0.833 | 0.817 | 0.808 |

Some example explanations for MNIST SuperPixel-Graph dataset are reported in Fig 5. Here we do not have results of GNNExplainer because the test model implemented by [36] for this dataset do not support the input of fractional adjacency matrix, which is an requirement for GNNExplainer to work. Instead, we compare PGM-Explainer with the SHAP extension for GNN and GRAD, a simple gradient approach. In this experiment, we do not restrict the number of nodes returned by PGM-Explainer. In fact, the nodes are included as long as they are in the Markov-blanket of the target variable. From that, we obtain explanations containing either $2, 3$, or $4$ nodes with an average of $3.08$. Since other methods do not have equivalent mechanism of selecting features, we set their number of returned nodes to 3. In GRAD, the top-3 nodes are chosen based on the sum gradients of the GNN's loss function with respect to the associated node features.

In our human-subjective test, we randomly chose 48 graphs whose predictions made by the GNN are correct. Then we use PGM-Explainer, SHAP, and GRAD to generate the corresponding explanations. Two examples of these explanations are provided in Fig 5(a) where nodes returned in explanation are highlighted in red boxes. Ten end-users were asked to give a score on a scale of 0 to 10 for each explanation based on the their thoughts on the importance of the highlighted nodes to the GNN prediction. From the distribution of the scores in Fig. 5(b), the explanations of PGM-Explainer are evaluated much higher than those of other methods. This means the nodes in the explanations generated by PGM-Explainer are more intuitive and might be more important to the GNN's predictions than those of other explanation methods.

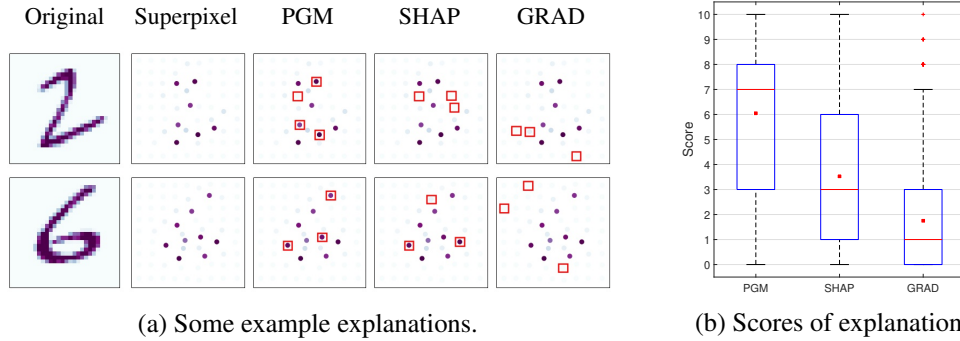

(a) Some example explanations.        (b) Scores of explanations.

Figure 5: Examples and distributions of scores of explanations generated by different explainers on the MNIST SuperPixel-Graph dataset. (a) The nodes returned in each explanations are highlighted in red boxes. (b) The distribution of scores given by end-users on the explanations. The red dots and red bars are the average and median values respectively.

## 5  Conclusion

In this paper, we propose PGM-Explainer, an explanation method faithfully explaining the predictions of any GNN in an interpretable manner. By approximating the target prediction with a graphical model, PGM-Explainer is able to demonstrate the non-linear contributions of explained features toward the prediction. Our experiments not only show the high accuracy and precision of PGM-Explainer but also imply that PGM explanations are favored by end-users. Although we only adopt Bayesian networks as interpretable models, our formulations of PGM-Explainer supports the exploration of others graphical models such as Markov networks and Dependency networks. For future researches, we would like to analyze the impact of different objective functions and structure learning methods on explainers' quality and the running time.

## Broader Impact

This paper proposed PGM-Explainer, which explains decisions of any GNN model in an interpretable manner, covering the non-linear dependency between features, a key aspect of graph data. Thus the contribution of this paper is fundamental and will have a broader impact on a vast number of applications, especially when many complex GNNs have been recently proposed and deployed in various domain fields. This paper will benefit a variety of high-impact GNNs based applications in terms of their interpretability, transparency, and fairness, including social network analysis, neural science, team science management, intelligent transportation systems, and critical infrastructures, to name a few.

## Acknowledgments and Disclosure of Funding

This work was supported in part by the National Science Foundation Program on Fairness in AI in collaboration with Amazon under award No. 1939725.

## Footnotes

[1]The proofs that the vanilla gradient-based explanation method and GNNExplainer fall into the class of *additive feature attribution methods* are provided in Appendix A.

[2]A scale-free network is a network whose degree distribution follows power law. The Barabási-Albert graph is a scale-free network generated by the Barabási–Albert model [31]

[3] Formal definition of Markov-blanket is provided in Appendix C.2

[4] Formal definition of a perfect map is provided in Appendix C.1

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
