[Supplementary Material]

# A    Additive feature attribution methods unify existing explainers for GNNs

In this section, we analyze the vanilla gradient-based explainers and GNNExplainer [24] under the explanation model framework. Our aim is to bring a clearer view on explanation models and show that the class of *additive feature attribution methods* introduced in [17] fully captures current explanation methods for GNNs.

**Gradient-based explainers.** The gradient-based approach has been one of the most popular explanation methods for conventional neural networks [18, 19, 40, 20, 21, 15]. Some of those recently have been analyzed in the context of GNNs [27], including Gradient-Based Saliency maps [15], Grad-CAM [18] and Excitation Back-Propagation [22]. All of these methods rely on back-propagating the GNN and assign important scores on explained features. Here, we consider the simplest gradient-based explanation method in which the score of each feature is associated with the gradient of the GNN's loss function with respect to that feature. The top-$M$ nodes with the largest sum of their features' score are selected as the explanation.

The proof that this explanation method falls into the class of *additive feature attribution methods* is quite straight-forward. Here, the interpretable domain $\mathcal{E}$ is a set of linear functions of the form $\zeta(z) = \phi_0 + \sum_{i=1}^{|V|} \phi_i z_i$, where $z_i \in \{0, 1\}$ is a binary variable representing the selection of node $i$ in the explanation and $\phi_i \in \mathbb{R}$ is the associated sum gradients in node $i$. The objective $R$ and the feasible sets $\mathcal{C}$ can be chosen as follows:

$$R_{\Phi,t}(\zeta) = \sum_{v \in V, z_v = 1} \left( \sum_{x \in X_v} \frac{\partial \Phi_t}{\partial x} \right), \quad \mathcal{C} = \left\{ \zeta(z) : \sum_{i=1}^{|V|} z_i = M \right\}. \tag{5}$$

To show other gradient-based explanation methods examined in [27] fall into the class of *additive feature attribution methods*, we just need to adjust the choice of the weight $\phi_i$ in the formulation.

**GNNExplainer.** To generate a subgraph $S = (V_S, E_S)$ with at most $M$ edges explaining a target prediction, GNNExplainer is formulated based on the following optimization:

$$\max_{S \subseteq G} \mathrm{I}(Y; S) = \max_{S \subseteq G} \left( \mathrm{H}(Y) - \mathrm{H}(Y|\mathcal{G} \equiv S) \right), \quad |E_S| \leq M, \tag{6}$$

where I and H are the mutual information function and the entropy function respectively. $S$ is a random subgraph of $G$ and $Y$ is a random variable with the probability that the target prediction belonging to each of $\mathcal{K}$ classes. In GNNExplainer, the distribution of $Y$ is obtained from the soft-output of the GNN at different input settings. The condition $\mathcal{G} \equiv S$ indicates that the realization of $\mathcal{G}$ must be consistent with the realization of subgraph $S$. The intuition on this objective is that, if knowing the information in the subgraph $S$ reduces the uncertainty of $Y$ significantly, the subgraph $S$ is a good explanation for the target prediction.

To avoid the complexity of directly solving (6), GNNExplainer relaxes the integer constraint on the entries of the input adjacency matrix $A$ and searches for an optimal fractional adjacency matrix $\tilde{A} \in [0, 1]^{|V| \times |V|}$. The distribution of $S$ is then defined by setting $P(S) = \prod_{(i,j) \in S} \tilde{A}_{ij}$. After that, by approximating $\mathrm{H}(Y|\mathcal{G} \equiv S) \approx \mathrm{H}(Y|\mathcal{G} \equiv \mathbb{E}[S]) = \mathrm{H}(Y|\tilde{A})$, the explainer uses mean-field optimization to solve for $\min_{\tilde{A}} \mathrm{H}(Y|\mathcal{G} \equiv \mathbb{E}[S])$. Finally, the explanation $S$ is selected based on the values of entries in the solution $\tilde{A}$. Note that in order to evaluate $\mathrm{H}(Y|\tilde{A})$, the explained model must be able to compute the prediction at fractional adjacency matrix input. However, in many current GNN's architecture such as GraphSage [6] and GIN-Graph [10], the information of the adjacency matrix is used to compute the aggregated discrete sum of features and operations on float inputs of adjacency matrix are not well-defined. Thus, GNNExplainer would fail to explain predictions of those models.

If we do not allow the entries of $\tilde{A}$ to receive fractional value, the optimization problem of GNNExplainer can be formulated under the *additive feature attribution methods* by setting $\zeta(\tilde{A}) = \phi_0 + \sum_{i=1}^{|V|} \sum_{j=1}^{|V|} \phi_{ij} \tilde{A}_{ij}$ where $\tilde{A}_{ij} \in \{0, 1\}$. The objective $R$ and the feasible set $\mathcal{C}$ are specified as follows:

$$R_{\Phi,t}(\zeta) = -\mathrm{H}(\Phi(\mathcal{G})_t|\tilde{A}), \quad \mathcal{C} = \left\{ \zeta(\tilde{A}) : \sum_{i=1}^{|V|} \sum_{j=1}^{|V|} \tilde{A}_{ij} \leq 2M, \tilde{A}_{ij} = \tilde{A}_{ji} \right\}. \tag{7}$$

The weights $\phi_{ij}$ of explanation model $\zeta$ are chosen based on the solution in maximizing $R_{\Phi,t}(\zeta)$.

# B    Example of PGM-Explainer with and without *no-child* constraint

In Fig. 6, we provide an example illustrating the impact of the *no-child* constraint (3) onto the PGM explanation. This experiment setup is the same as that in experiment of Fig. 1. Here, we use PGM-Explainer to explain the prediction on role "blue" of node $C$ shown in Fig.6a. Fig.6b and Fig.6c are the resulted explanations without and with the *no-child* constraint.

(a) Ground-truth motif.      (b) Without *no-child* constraint.      (c) With *no-child* constraint.

Figure 6: Example of PGM-explanation with and without *no-child* constraint. **(a)** The ground-truth motif containing the target prediction to be explained, which is the prediction on node $C$. **(b)** A PGM-explanation in a form of Bayesian network without the *no-child* constraint. **(c)** A PGM-explanation in a form of Bayesian network with the *no-child* constraint.

As can be seen in this experiment, imposing the constraint does not change the set of nodes identified by the explainer. However, the constraint changes the edges in the Bayesian network. In general, without the constraint, we will obtain a more simple model: less edges, less number of parameters and more independence assertions. While this network is faithful to the data, it might be misleading to non-experts. In fact, the Markov-blanket of $C$ in this graph is $\{A, B, E\}$. Notice that $E$ is in the Markov blanket even when there is no chain of one-direction arrows connecting $C$ and $E$. By imposing the *no-child* constraint, the resulted network here reverses the arrow $C \to A$ and add two additional arrows $B \to A$ and $E \to C$. These arrows are included to guarantee that the resulted network has enough power to represent the distribution generating the data. Even though this network is more complex, it is intuitive in the sense that we can directly point out the Markov-blanket of $C$, which is its parents. Furthermore, if the Bayesian network is stored in factorization form, it is also much more convenient to perform inference queries on the target variable $C$ when it has no child. To see this, consider the factorize forms of the networks in Fig. 6b and 6c, which are $P(B)P(C|B)P(D|B)P(E|D)P(A|C,E)$ and $P(B)P(D|B)P(E|D)P(A|B,E)P(C|A,B,E)$ respectively. When storing the networks, we only store each conditional probability. Thus, answering inference query regarding the target variable $C$ is much more convenient in the latter network as all required information is stored in $P(C|A,B,E)$. To sum up, the *no-child* constraint might increase the complexity of the explanation model; however, it offers better intuition and more convenient in performing inference on the target of explanation.

# C   Probabilistic Graphical Model preliminaries

For completeness of this work, we provide some formal definitions of PGM-related concepts in this section. For a more and complete information about PGM, readers are referred to [28].

## C.1   I-map and Perfect map

Given a distribution $P$, we denote $\mathcal{I}(P)$ to be the set of independence assertions of the form $(\boldsymbol{x} \perp\!\!\!\perp \boldsymbol{y}|\boldsymbol{z})$ that hold in $P$. Similarly, given a Bayesian network $\mathcal{B}$, $\mathcal{I}(\mathcal{B})$ is the set of independence assertions made by $\mathcal{B}$. For $\mathcal{B}$ to faithfully represent $P$, $\mathcal{B}$ must not contain any independence assertions that are not made by $P$, i.e. $\mathcal{I}(\mathcal{B}) \subseteq \mathcal{I}(P)$. In that case, we say $\mathcal{B}$ is an I-map for $P$.

In the context of PGM, we normally want to find the simplest model $\mathcal{B}$ that is able to represent $P$. Roughly speaking, the larger the set $\mathcal{I}(\mathcal{B})$, the simpler the model $\mathcal{B}$ since we would need less number of parameters to determine the network. This leads us to the following definition of minimal I-map.

**Definition 1** *A Bayesian network $\mathcal{B}$ is a minimal I-map for $P$ if it is an I-map for $P$, and if the removal of even a single edge from $\mathcal{B}$ renders it not an I-map.*

Therefore, to capture the independence structure in $P$, a standard solution is to search for minimal I-maps of $P$. However, as shown by [28], even if a network $\mathcal{B}$ is a minimal I-map, it may not include many important independence assertions in $P$ and fails to capture the distribution's structure. Alternatively, we aim to find a graph, called perfect map, that precisely capture $P$.

**Definition 2** *A Bayesian network $\mathcal{B}$ is a perfect map for $P$ if $\mathcal{I}(\mathcal{B}) = \mathcal{I}(P)$.*

Unfortunately, not every distribution $P$ has a perfect map. In the scope of this work, we consider the case when the perfect map for the distribution generating the data exists, i.e. the class of Bayesian networks has the power to represent the distribution precisely.

## C.2   Markov-blanket

In the variable selection step of PGM-Explainer (Section 3.2), we solve for the set $\boldsymbol{U}(\boldsymbol{t})$ that guarantees to contain the Markov-blanket of the target variable in case the perfect map exists. Here, we provide the formal definition of the Markov-blanket in a distribution.

**Definition 3** *Given a distribution $P$ on a set of random variables $\mathcal{X}$, a set $\boldsymbol{S}$ is a Markov-blanket of $\boldsymbol{x} \in \mathcal{X}$ if $\boldsymbol{x} \notin \boldsymbol{S}$ and if $\boldsymbol{S}$ is a minimal set such that $(\boldsymbol{x} \perp\!\!\!\perp \mathcal{X} \setminus (\{\boldsymbol{x}\} \cup \boldsymbol{S})|\boldsymbol{S}) \in \mathcal{I}(P)$. We denote this set $\mathrm{MB}_{\mathcal{P}}(\boldsymbol{x})$.*

If the perfect map $\mathcal{B}^*$ for $P$ exists, then $\mathrm{MB}_{\mathcal{P}}(\boldsymbol{x})$ is simply $\boldsymbol{x}$'s parents, $\boldsymbol{x}$'s children, and $\boldsymbol{x}$'s children's other parents in $\mathcal{B}^*$ [28].

## C.3   Structure learning and Bayesian Information Criterion (BIC) score

The task of learning a graph structure from data is called structure learning. Algorithms for structure learning can be roughly classified into two categories: score-based and constraint-based. The score-based algorithms associate a score to each candidate structure with respect to the training data, and then search for a high-scoring structure. The constraint-based algorithms look for dependencies and conditional dependencies in the data and construct the structure accordingly.

In score-based approach, the selection of scoring function is critical. A natural choice of the score function is the likelihood $l(\hat{\theta}_{\mathcal{B}} : \mathcal{D})$, where $\mathcal{D}$ is the data, $\mathcal{B}$ is the graphical model of consideration, $\hat{\theta}_{\mathcal{B}}$ is the maximum likelihood parameters for $\mathcal{B}$ and $l$ is the logarithm of the likelihood function. However, as pointed out in [28], the likelihood score overfits the training data and returns overly complex structure. An alternative solution is the Bayesian score

$$\mathrm{score}_B(\mathcal{B} : \mathcal{D}) := \log P(\mathcal{D}|\mathcal{B}) + \log P(\mathcal{B}), \qquad (8)$$

where $P(\mathcal{B})$ is the prior over structures. However, this prior is almost insignificant compared to the marginal likelihood $P(\mathcal{D}|\mathcal{B})$, which can be computed as:

$$P(\mathcal{D}|\mathcal{B}) = \int P(\mathcal{D}|\theta_{\mathcal{B}}, \mathcal{B})P(\theta_{\mathcal{B}}|\mathcal{B})d\theta_{\mathcal{B}}. \qquad (9)$$

It can be shown that, if we use a Dirichlet parameter prior for all parameters in $\mathcal{B}$, then, when the number of samples $n$ is sufficiently large, we have

$$\log P(\mathcal{D}|\mathcal{B}) = l(\hat{\theta}_{\mathcal{B}} : \mathcal{D}) - \frac{\log n}{2}\text{Dim}[\mathcal{B}] + O(1). \tag{10}$$

Thus, the *BIC score* is defined as

$$\text{score}_{BIC}(\mathcal{B} : \mathcal{D}) := l(\hat{\theta}_{\mathcal{B}} : \mathcal{D}) - \frac{\log n}{2}\text{Dim}[\mathcal{B}] \tag{11}$$

which can be considered an approximation of the Bayesian score when the data is sufficiently large.

In PGM-Explainer, we use *BIC score* for our structure learning step. One main reason is *BIC score* is known to be *consistent*. The definition of a score to be *consistent* is given as follows.

**Definition 4** *Assume distribution $P$ has a perfect map $\mathcal{B}^*$. We say that a scoring function is consistent if the following properties hold as the amount of data $n \to \infty$, with probability that approaches 1:*

1. *The structure $\mathcal{B}^*$ will maximize the score.*

2. *All structure $\mathcal{B}$ such that $\mathcal{I}(\mathcal{B}) \neq \mathcal{I}(\mathcal{B}^*)$ will have strictly lower score.*

This condition implies that, for all structures containing different set of independence assertions from those in the distribution (or in the perfect map), they will have strictly lower *BIC score*. Hence, by maximizing *BIC score*, PGM-Explainer theoretically searches for the structure containing all and only the dependence assertions encoded in the data.

## D  Proof of Theorem 1

This section provides the proof of Theorem 1, Section 3.2.

**Proof 1** *Let's consider a node $v' \in \text{MB}_{\mathcal{B}^*}(t)$. In Bayesian network $\mathcal{B}^*$, the Markov-blanket of $t$ is the union of $t$'s parents, $t$'s children, and $t$'s children's other parents [28]. In other words, at least one of the three followings cases must holds:*

1. *$v' \in \text{Pa}_{\mathcal{B}^*}(t)$.*

2. *$v' \in \text{Ch}_{\mathcal{B}^*}(t)$.*

3. *$\exists v$ such that $v' \in \text{Pa}_{\mathcal{B}^*}(v)$ and $t \in \text{Pa}_{\mathcal{B}^*}(v)$.*

*For the first two cases, either $v' \to t$ or $t \to v'$ is an active path of $\mathcal{B}^*$. As $\mathcal{B}^*$ is a perfect map for $P$, we have $v' \not\perp t$ and $v'$ must be included in $S(t)$. Since $v \in S(v)$ for all $v$, we have $S(t) \subseteq \cup_{v \in S(t)} S(v)$ and $v' \in \cup_{v \in S(t)} S(v)$.*

*For the last case, as $v' \in \text{Pa}_{\mathcal{B}^*}(v)$ and $t \in \text{Pa}_{\mathcal{B}^*}(v)$, we have $v' \to v$ and $t \to v$ are active paths of $\mathcal{B}^*$. Thus, $v' \not\perp v$ and $v \not\perp t$. Consequently $v'$ must be included in $S(v)$ where $v \in S(t)$, i.e. $v' \in \cup_{v \in S(t)} S(v)$.*

## E  Proof of Theorem 2

This section provides the proof of Theorem 2, Section 3.2.

**Proof 2** *Since $t$ has no child, $\text{MB}_{\mathcal{B}^*}(t) = \text{Pa}_{\mathcal{B}^*}(t)$ where $\text{Pa}_{\mathcal{B}^*}(t)$ is the set of parents of $t$ in $\mathcal{B}^*$. Hence, for any node $v \in \text{MB}_{\mathcal{B}^*}(t)$, $v \in \text{Pa}_{\mathcal{B}^*}(t)$ and $v \to t$ is an active path of $\mathcal{B}^*$. As $\mathcal{B}^*$ is a perfect map for $P$, we have $v \not\perp t$ and $v$ must be included in $S(t)$.*

# F PGM-Explainer without *no-child* constraint

---

**Algorithm 1:** Generating PGM explanation for GNN without no-child constraint

---

**Input** : Model $\Phi$, input graph $G$, target $t$, size constraint $M$, and sampling size $n$.

**Output** : A Bayesian network $\mathcal{B}_t$ explains prediction $\Phi(G)_t$

**Step 1:** *Data generation*

    $L$ = the number of GNN layers of $\Phi$.

    Get the $L$-hop neighbor graph $\text{Ne}_t^G$

    Initiate an empty data $\mathcal{D}_t$

    For $i = 1$ to $n$ do:

        Generate random $\boldsymbol{s} \in \{0,1\}^{|\text{Ne}_t^G|}$

        Compute realization graph $G(\boldsymbol{s})$

        Compute prediction $\Phi(G(\boldsymbol{s}))$

        For each $v \in \text{Ne}_t^G$, compute $\boldsymbol{v} = \{s_v, \text{I}(\Phi(G(\mathbf{s}))_v)\}$

        Record $\boldsymbol{v}$ into the entry of node $v$ in data $\mathcal{D}_t$.

    End

**Step 2:** *Generate* $\boldsymbol{U(t)}$

    Initiate an empty set $\boldsymbol{U(t)}$

    For each $v \in \text{Ne}_t^G$, if $\boldsymbol{v} \not\perp\!\!\!\perp \boldsymbol{t}$, add $v$ to $S(\boldsymbol{t})$.

    For each $v \in S(\boldsymbol{t})$:

        For each $v' \in \text{Ne}_t^G \setminus \{v\}$ if $\boldsymbol{v}' \not\perp\!\!\!\perp \boldsymbol{v}$, add $v'$ to $U(\boldsymbol{t})$.

    Rank the nodes in $\boldsymbol{U(t)}$ based on their dependencies with $\boldsymbol{t}$.

    Keep the top $M$ dependent variables in $\boldsymbol{U(t)}$.

**Step 3:** *Structure Learning - Generate the explanation* $\hat{\mathcal{B}}$:

    $\hat{\mathcal{B}} = \arg\max_{\mathcal{B}} \text{score}_{BIC}(\mathcal{B} : \mathcal{D}_t[\boldsymbol{U(t)}])$

    Return $\hat{\mathcal{B}}$.

---

# G PGM-Explainer with *no-child* constraint

---

**Algorithm 2:** Generating PGM explanation for GNN with no-child constraint

---

**Input** : Model $\Phi$, input graph $G$, target $t$, size constraint $M$, and sampling size $n$.

**Output** : A Bayesian network $\mathcal{B}_t$ explain the prediction $\Phi(G)_t$

**Step 1:** *Data generation*

    (The same as in Step 1 of Algorithm 1, Appendix F)

**Step 2:** *Generate* $\boldsymbol{U(t)}$

    Initiate an empty set $\boldsymbol{U(t)}$

    For each $v \in \text{Ne}_t^G$, if $\boldsymbol{v} \not\perp\!\!\!\perp \boldsymbol{t}$, add $v$ to $\boldsymbol{U(t)}$.

    Rank the nodes in $\boldsymbol{U(t)}$ based on their dependencies with $\boldsymbol{t}$.

    Keep the top $M$ dependent variables in $\boldsymbol{U(t)}$.

**Step 3:** *Structure Learning - Generate the explanation* $\hat{\mathcal{B}}$:

    $\hat{\mathcal{B}}' = \arg\max_{\mathcal{B}'} \text{score}_{BIC}(\mathcal{B}' : \mathcal{D}_t[\boldsymbol{U(t)} \setminus \{\boldsymbol{t}\}])$

    Adding $\boldsymbol{t}$ to $\hat{\mathcal{B}}'$ and obtain $\hat{\mathcal{B}}$.

    Initiate set $\text{Pa}(\boldsymbol{t}) = \boldsymbol{U(t)}$

    While $\exists \boldsymbol{v} \in \text{Pa}(\boldsymbol{t})$ such that $\boldsymbol{v} \perp\!\!\!\perp \boldsymbol{t} | \text{Pa}(\boldsymbol{t}) - \{\boldsymbol{v}\}$, remove $\boldsymbol{v}$ from $\text{Pa}(\boldsymbol{t})$.

    For all $\boldsymbol{v} \in \text{Pa}(\boldsymbol{t})$, add edge $(\boldsymbol{v}, \boldsymbol{t})$ to $\hat{\mathcal{B}}$.

    Return $\hat{\mathcal{B}}$.

---

## H   Proof of Theorem 3

This section provides the proof of Theorem 3, Section 3.3.

**Proof 3** *From proposition 18.1 [28], we have the log-likelihood between a data $\mathcal{D}$ on random variables $V$ and the model $\mathcal{B}$ with parameters $\hat{\theta}_{\mathcal{B}}$, $l(\hat{\theta}_{\mathcal{B}} : \mathcal{D})$, can be rewritten as follows*

$$l(\hat{\theta}_{\mathcal{B}} : \mathcal{D}) = n \sum_{\boldsymbol{v} \in V} \boldsymbol{I}_{\hat{P}}(\boldsymbol{v}; \mathrm{Pa}_{\mathcal{B}}(\boldsymbol{v})) - n \sum_{\boldsymbol{v} \in V} \mathrm{H}_{\hat{P}}(\boldsymbol{v}) \tag{12}$$

*where $\hat{P}$ is the empirical distribution.*

*We now consider the first case where the output $\hat{\mathcal{B}}$ of Algorithm 2 implies an independence assumption that $\mathcal{B}^*$ does not support. Thus, the set of independence assumptions of $\hat{\mathcal{B}}$ is not contained in that set in $P$ and we say, by the definition of I-map [28], $\hat{\mathcal{B}}$ is not an I-map of $P$. Therefore, the log-likelihood $l(\hat{\theta}_{\hat{\mathcal{B}}} : \mathcal{D}_t)$ is less than $l(\hat{\theta}_{\mathcal{B}^*} : \mathcal{D}_t)$ and we have*

$$\sum_{\boldsymbol{v} \in V} \boldsymbol{I}_P(\boldsymbol{v}; \mathrm{Pa}_{\mathcal{B}^*}(\boldsymbol{v})) > \sum_{\boldsymbol{v} \in V} \boldsymbol{I}_P(\boldsymbol{v}; \mathrm{Pa}_{\hat{\mathcal{B}}}(\boldsymbol{v})) \tag{13}$$

*where $V$ is the set of nodes of $\mathcal{B}^*$.*

*Since the sets of parents of $\boldsymbol{t}$ in $\mathcal{B}^*$ and that set in $\hat{\mathcal{B}}$ are the same, we have*

$$\sum_{\boldsymbol{v} \in V \setminus \{\boldsymbol{t}\}} \boldsymbol{I}_P(\boldsymbol{v}; \mathrm{Pa}_{\mathcal{B}^*}(\boldsymbol{v})) + \boldsymbol{I}_P(\boldsymbol{t}; \mathrm{Pa}_{\mathcal{B}^*}(\boldsymbol{t})) > \sum_{\boldsymbol{v} \in V \setminus \{\boldsymbol{t}\}} \boldsymbol{I}_P(\boldsymbol{v}; \mathrm{Pa}_{\hat{\mathcal{B}}}(\boldsymbol{v})) + \boldsymbol{I}_P(\boldsymbol{t}; \mathrm{Pa}_{\hat{\mathcal{B}}}(\boldsymbol{t})) \tag{14}$$

$$\Rightarrow \sum_{\boldsymbol{v} \in V \setminus \{\boldsymbol{t}\}} \boldsymbol{I}_P(\boldsymbol{v}; \mathrm{Pa}_{\mathcal{B}^*}(\boldsymbol{v})) > \sum_{\boldsymbol{v} \in V \setminus \{\boldsymbol{t}\}} \boldsymbol{I}_P(\boldsymbol{v}; \mathrm{Pa}_{\hat{\mathcal{B}}}(\boldsymbol{v})) \tag{15}$$

*We denote $\mathcal{B}'^*$ the Bayesian network obtained by removing node $\boldsymbol{t}$ and the related edges from $\mathcal{B}^*$. Since $\boldsymbol{t}$ has no child in $\mathcal{B}^*$, $\mathrm{Pa}_{\mathcal{B}^*}(\boldsymbol{v}) = \mathrm{Pa}_{\mathcal{B}'^*}(\boldsymbol{v})$ for all $\boldsymbol{v} \in V \setminus \{\boldsymbol{t}\}$. Thus, we have*

$$\sum_{\boldsymbol{v} \in V \setminus \{\boldsymbol{t}\}} \boldsymbol{I}_P(\boldsymbol{v}; \mathrm{Pa}_{\mathcal{B}^*}(\boldsymbol{v})) = \sum_{\boldsymbol{v} \in V \setminus \{\boldsymbol{t}\}} \boldsymbol{I}_P(\boldsymbol{v}; \mathrm{Pa}_{\mathcal{B}'^*}(\boldsymbol{v})) \tag{16}$$

*Similarly, as $\boldsymbol{t}$ has no child in $\hat{\mathcal{B}}$, $\mathrm{Pa}_{\hat{\mathcal{B}}}(\boldsymbol{v}) = \mathrm{Pa}_{\hat{\mathcal{B}}'}(\boldsymbol{v})$ for all $\boldsymbol{v} \in V \in V \setminus \{\boldsymbol{t}\}$ and the following holds*

$$\sum_{\boldsymbol{v} \in V \setminus \{\boldsymbol{t}\}} \boldsymbol{I}_P(\boldsymbol{v}; \mathrm{Pa}_{\hat{\mathcal{B}}}(\boldsymbol{v})) = \sum_{\boldsymbol{v} \in V \setminus \{\boldsymbol{t}\}} \boldsymbol{I}_P(\boldsymbol{v}; \mathrm{Pa}_{\hat{\mathcal{B}}'}(\boldsymbol{v})) \tag{17}$$

*Combining (15), (16) and (17), we have*

$$\sum_{\boldsymbol{v} \in V \setminus \{\boldsymbol{t}\}} \boldsymbol{I}_P(\boldsymbol{v}; \mathrm{Pa}_{\mathcal{B}'^*}(\boldsymbol{v})) > \sum_{\boldsymbol{v} \in V \setminus \{\boldsymbol{t}\}} \boldsymbol{I}_P(\boldsymbol{v}; \mathrm{Pa}_{\hat{\mathcal{B}}'}(\boldsymbol{v})) \tag{18}$$

*This leads us to the following results on the BIC score of $\mathcal{B}'^*$ and $\hat{\mathcal{B}}'$ as $n \to \infty$:*

$$\mathrm{score}_{BIC}(\mathcal{B}'^* : \mathcal{D}'_t) - \mathrm{score}_{BIC}(\hat{\mathcal{B}}' : \mathcal{D}'_t) \tag{19}$$

$$= l(\hat{\theta}_{\mathcal{B}'^*} : \mathcal{D}'_t) - l(\hat{\theta}_{\hat{\mathcal{B}}'} : \mathcal{D}'_t) - \frac{\log n}{2} \left( \mathrm{Dim}[\mathcal{B}'^*] - \mathrm{Dim}[\hat{\mathcal{B}}'] \right) \tag{20}$$

$$= n \left( \sum_{\boldsymbol{v} \in V \setminus \{\boldsymbol{t}\}} \boldsymbol{I}_{\hat{P}}(\boldsymbol{v}; \mathrm{Pa}_{\mathcal{B}'^*}(\boldsymbol{v})) - \sum_{\boldsymbol{v} \in V \setminus \{\boldsymbol{t}\}} \boldsymbol{I}_{\hat{P}}(\boldsymbol{v}; \mathrm{Pa}_{\hat{\mathcal{B}}'}(\boldsymbol{v})) \right) - \frac{\log n}{2} \left( \mathrm{Dim}[\mathcal{B}'^*] - \mathrm{Dim}[\hat{\mathcal{B}}'] \right) \tag{21}$$

$$\approx n\Delta - \frac{\log n}{2} \left( \mathrm{Dim}[\mathcal{B}'^*] - \mathrm{Dim}[\hat{\mathcal{B}}'] \right) \tag{22}$$

*where $\Delta = \sum_{\boldsymbol{v} \in V \setminus \{\boldsymbol{t}\}} \boldsymbol{I}_P(\boldsymbol{v}; \mathrm{Pa}_{\mathcal{B}'^*}(\boldsymbol{v})) - \sum_{\boldsymbol{v} \in V \setminus \{\boldsymbol{t}\}} \boldsymbol{I}_P(\boldsymbol{v}; \mathrm{Pa}_{\hat{\mathcal{B}}'}(\boldsymbol{v}))$. The last approximation (22) is from the fact that, as $n \to \infty$, the empirical distribution $\hat{P}$ converge to $P$. Since $\Delta > 0$ by (18)*

*the expression in (22) approaches positive infinity when $n \to \infty$ and we have $\text{score}_{BIC}(\mathcal{B}'^* : \mathcal{D}_t') >$*
*$\text{score}_{BIC}(\hat{\mathcal{B}}' : \mathcal{D}_t')$. This is a contradiction to the fact that $\hat{\mathcal{B}}'$ maximizes $\text{score}_{BIC}(\mathcal{B}' : \mathcal{D}_t')$.*

*We now consider the remain case where the independence assumptions in $\hat{\mathcal{B}}$ contains all the independence assumptions in $\mathcal{B}^*$, and $\mathcal{B}^*$ contains an independence assumption that $\hat{\mathcal{B}}$ does not. Thus, $\hat{\mathcal{B}}$ is able to represent any distributions that $\mathcal{B}^*$ can. As $\hat{P}$ converges to $P$ when $n \to \infty$, we have the log-likelihood of the two networks are the equal*

$$l(\hat{\theta}_{\hat{\mathcal{B}}} : \mathcal{D}_t) = l(\hat{\theta}_{\mathcal{B}^*} : \mathcal{D}_t) \tag{23}$$

*Using similar tricks as in (15), (16) and (17) for the previous case, we will obtain*

$$\sum_{\boldsymbol{v} \in V \setminus \{\boldsymbol{t}\}} \boldsymbol{I}_P(\boldsymbol{v}; \text{Pa}_{\mathcal{B}'^*}(\boldsymbol{v})) = \sum_{\boldsymbol{v} \in V \setminus \{\boldsymbol{t}\}} \boldsymbol{I}_P(\boldsymbol{v}; \text{Pa}_{\hat{\mathcal{B}}'}(\boldsymbol{v})) \tag{24}$$

*Now, the difference between the BIC scores of $\mathcal{B}'^*$ and $\hat{\mathcal{B}}'$ as $n \to \infty$ is*

$$\text{score}_{BIC}(\mathcal{B}'^* : \mathcal{D}_t') - \text{score}_{BIC}(\hat{\mathcal{B}}' : \mathcal{D}_t') \tag{25}$$

$$= l(\hat{\theta}_{\mathcal{B}'^*} : \mathcal{D}_t') - l(\hat{\theta}_{\hat{\mathcal{B}}'} : \mathcal{D}_t') - \frac{\log n}{2} \left( \text{Dim}[\mathcal{B}'^*] - \text{Dim}[\hat{\mathcal{B}}'] \right) = \frac{\log n}{2} \left( \text{Dim}[\hat{\mathcal{B}}'] - \text{Dim}[\mathcal{B}'^*] \right) \tag{26}$$

*As $\hat{\mathcal{B}}$ has less independence assumptions than $\mathcal{B}^*$, $\text{Dim}[\hat{\mathcal{B}}] - \text{Dim}[\mathcal{B}^*] > 0$. Furthermore, the reduction in number of parameters from $\hat{\mathcal{B}}$ to $\hat{\mathcal{B}}'$ is the same as the reduction from $\mathcal{B}^*$ to $\mathcal{B}'^*$ (remove one node and the same set of edges). Thus we have $\text{Dim}[\hat{\mathcal{B}}'] - \text{Dim}[\mathcal{B}'^*] > 0$. This constitutes a contradiction in the assumption that $\hat{\mathcal{B}}'$ maximizes $\text{score}_{BIC}(\mathcal{B}' : \mathcal{D}_t')$.*

## I  Parameters of synthetic dataset

Table 4: Parameters of synthetic datasets.

| Dataset | Base | Motifs | Node's Features |
|---------|------|--------|-----------------|
| Syn 1 | 300-node BA graph | 80 5-node house-shaped motif | Constant |
| Syn 2 | 350-node BA graph | 100 5-node house-shaped motif | Generated from Labels |
| Syn 3 | 300-node BA graph | 80 9-node grid-shaped motif | Constant |
| Syn 4 | Tree with height 8 | 60 6-node cycle-shaped motif | Constant |
| Syn 5 | Tree with height 8 | 80 9-node grid-shaped motif | Constant |
| Syn 6 | 300-node BA graph | 80 5-node bottle-shaped motif | Constant |