[Reviews · NeurIPS 2020]

Review 1

Summary and Contributions: The paper propose a Bayesian approach for building a model-agnostic explainer for graph neural networks. The essential contribution of the paper is to explore an alternative method to the additive feature attribution explainer family. This alternative is a probabilistic graphical model explainer which doesn't rely on the assumption of linear independence of explained features.

Strengths: The claims are well supported by a strong mathematical background. The variables selection step of the pipeline addressed cleverly the task of reducing the number of variables and choosing the important ones by a using a very interesting approaches based on the Markov-blanket properties. The results are interesting - showing some advancements.

Weaknesses: I'm questioning the choice of graph perturbation, why this choice? What are alternatives ?

Correctness: The claims and method are correct and theoretically well grounded. No obvious errors are observable in the methodology.

Clarity: The contributions of this works and the issues of previous works are clearly identified. The text is clear and logically organized in order to provide the appropriated explanations of almost all potential questions.

Relation to Prior Work: Related work is relevant and appropriate.

Reproducibility: Yes

Additional Feedback: - At equation (2) M should be defined. - The concept of Barabasi-Albert graph should be explained - The notion of structure notion should be defined when used the first time. - A minimal explanation of Bayesian Networks should be included in the preliminaries with minimal and specific references - Could you give a formal definition of the Markov-Blanket? - What is definition of perfect map ? - Could be good idea to give the meaning of symbol used (in order make the reading easier) those used for conditional independence , definition, likelihood. - Define the BIC score (Bayesian information criterium) and give an overall signification to grasp is meaning. (However, I have appreciated to read what were the reasons for choosing) - Why saying BIC score should reflect ... ?It's possible to demonstrate it asymptotically ? - SHAP, define it. (shapley additive explanation) - Why GNN must integrate network features in a non-linear manner ? - It's not clear what is the blue values in the table 1 (GNNExplainer) - What is superpixel graph ? Minimal explanation in order understand the figure 5a ** Acknowledge of authors feedback


Review 2

Summary and Contributions: This work aims at generating explanations for graph neural networks based on Bayesian network and proposes a Probabilistic Graphical Model (PGM) model-agnostic explainer for GNNs, which provides statistical information on the contributions of graph’s components in terms of conditional probabilities. It proposes a new framework according to PGM so as to illustrate the dependencies of underlying graph components with regard to the target variable. In order to provide an efficient and tractable algorithm, this work introduces several simplified tricks and provides theoretical analysis on the statistical information that can be encoded. Furthermore, experiments on both synthetic and real-world datasets demonstrate that PGM-Explainer can provide more accurate and intuitive explanations for GNN predictions. As for contributions, this work adopts Bayesian networks as GNN interpretable models, aiming at explaining GNN decisions in a new interpretable manner. This manuscript also demonstrates its explanation power empirically and reveal its potential applications on various domains.

Strengths: Different from existing gradient based or mutual information based explainers, this work provides PGM-Explainer, a dependency based explanation model and verify its power. (Section 4) Experimental results show that PGM-Explainer consistently outperforms other methods included in comparison on synthetic and real-world data. PGM-Explainer also can capture more explainable relationships/dependences.

Weaknesses: (Section 4) The complexity of model and its training were not analyzed. Comparison of the running time among 3 different explainers can help. Without theoretical and empirical analysis, it is unclear how applicable PGM-Explainer is to wider range of applications. (Figure 5 around Line 311) It is expected to provide more explanation about the results as well as more comprehensive results (e.g., from different number of selected nodes), which can be added in the appendix.

Correctness: Yes.

Clarity: The arrangement of the paper is in good order. It first introduces the necessity of explanation models, the overall framework for explainers and then proposes PGM-Explainers in a consistent way. Several illustrative examples give intuitive introduction, which is easy for understanding. See Figure 1 for the explanation of a GNN’s prediction and Figure 5 for graph classification explanations generated by different explainers. There are numerous complex notations and thus hinders readers from straightforward understanding. It takes long time to remember the meaning of notations and sometimes leads to confusion and ambiguity.

Relation to Prior Work: This paper provides detailed description of different explainers, providing a clear view on the differences among gradient based, mutual information based and dependency based explanation models. It conducts comparisons on a wide range of datasets including synthetic data and real-world data, ranging from node classification tasks to graph classification tasks. However, this work doesn’t make comparison with popular attention based GNNs.

Reproducibility: Yes

Additional Feedback: It is better to present some intuitive and simple description before detailedmathematical description.


Review 3

Summary and Contributions: This paper proposes a novel interpretation method, PGM-explainer, to provide an explanation of a Graph Neural Network (GNN) prediction in the form of a probabilistic graph model (PGM). With PGM-explainer, they provide not only highly contributing nodes in an input graph but also the dependencies among the nodes using conditional probabilities. PGM-explainer comprises three stages; data generation for perturbing input graph, variable selection for improving computational efficiency and compactness, and structure learning for building PGM (Bayesian network). They gave the theoretical proof that generated PGM always includes the Markov-blanket of the target prediction with a Bayesian network. They experimented on synthetic and real-world datasets and showed superior accuracy (or precision) and end-user faithfulness compared with the existing methods. ** post-rebuttal update *** I have read the author rebuttal. I think that the authors tried their best to address the concerns raised by the reviewers and appreciate their efforts.

Strengths: PGM-explainer is a novel interpretation method which leverages mathematically grounded Bayesian network. Computation could have been very burdensome, but they effectively reduced the computational complexity by introducing various theorem which is thoroughly proven. Since the proposed method does not rely on the linear independence assumption, it provides deeper explanations including dependency among features. As a result, the proposed method can cover more flexible distributions of graph structures and variables compared to existing methods. The conditional probabilities in Figure 1 seem very helpful for deeply understanding the GNNs’ prediction.

Weaknesses: The reported accuracies and precisions only prove that the highly contributed nodes appear in the explanation. There is no verification of the induced dependencies. Some of their arguments are not grounded concretely. If they cannot provide the justification, it would be better to smooth the expressions (e.g. the argument that the mutual information of GNNExplainer is not appropriate for many nodes is not concrete (line 302-303).) In addition, the description of experimental settings seems not enough to understand an implementation. It should provide more details, for example, an initial embeddings and model depth (although same as the setting from previous works). Moreover, the paper should report more ablation study results with various experimental settings to verify the underlying intuitions and theoretical analyses.

Correctness: I did not find any questionable points in the manuscript.

Clarity: This paper is written easily and clearly considering the complex theorems and methodologies.

Relation to Prior Work: Highly related to the prior work.

Reproducibility: Yes

Additional Feedback: In order to address the concerns in the weaknesses, Please verify the validity of the dependencies in an explanation. Please provide the justification line 302-303, which argues that the mutual information used in GNNExplainer is not suitable for the large number of nodes. Please give the explanation about why the Laplacian-based GCN ([5] Kipf and Welling, 2017) was picked to train MNIST SuperPixel-Graph dataset (among various types of existing GNNs).


Review 4

Summary and Contributions: This paper proposes a model-agnostic explanation algorithm, called PGM-Explainer, for graph neural networks by identifying important graph components for a given target. PGM-Explainer can illustrate the dependency among the explained features by searching an optimal structure for the filtered important neighbors. Evaluations on the synthetic and real-world datasets show the effectiveness of PGM-Explainer.

Strengths: 1. The writing is good and easy to read. Examples are given before the authors illustrate the details of the proposed algorithm. 2. The tackled problem is of significance. GNN explanation is an important and promising topic with the burst of GNN models these days. Existing explainable GNN algorithms are all interpretable models that can make predictions as well as provide the evidence for the final predictions. However, PGM-Explainer is a model-agnostic algorithm to explain the predictions from black-box GNN predictors. Thus, the black-box GNN predictors can be a simple and efficient model with high prediction accuracy. 3. The proposed algorithm is reasonable. The data generation step is to generate a set of input neighbors and output prediction pairs for explanation. The variable selection is to select important variables for the final explanation. The last step is to learn the relationship between the selected important variables.

Weaknesses: 1. What is the black-box GNN model used in the experiments? What is the prediction accuracy of the GNN model? 2. The paper lacks the time complexity analysis for the proposed algorithm, especially for the first step of the entire algorithm. How many trials are used to generate perturbed feature series?

Correctness: Yes

Clarity: Yes

Relation to Prior Work: Yes

Reproducibility: No

Additional Feedback: The author's rebuttal has addressed by concern.

[Author Response · NeurIPS 2020]

We would like to thank reviewers for the constructive feedback and the positive recommendation. The reference
numbers used here are from the manuscript. Following are our responses to the concerns raised by reviewers.

**(R1) The choice of graph perturbation method.** There is a trade-off in the amount of perturbation applied on the
input graph: too large perturbation makes the explainer fail to capture the local behavior of the model at the input, while
too small perturbation prevents the explainer from efficiently observing the contributions of graph's components toward
the model's output. While zeroing-out features leads to abrupt change in the model's output, randomizing features
requires a large number of samples for stable explanations. We observe that averaging features among nodes lead to
better performance for PGM-Explainer. As stated at line 140, this method is used in our implementation.

**(R2) $\perp\!\!\!\perp$ symbol in Alg.1.** Thank you for pointing this out! It should be $\not\perp\!\!\!\perp$ which indicates independent relationship.

**(R2) 4 nodes in Fig. 5.** The 4 nodes highlighted in Fig. 5 of PGM-Explainer is not a mistake. In this experiment,
we do not restrict the number of nodes returned by PGM-Explainer. In fact, they are included as long as they are in
the Markov-Blanket of the target variable. From that, we obtain explanations containing either $2, 3$, or $4$ nodes with
an average of $3.08$. Since other methods such as SHAP and GRAD do not have equivalent mechanism of selecting
features, we set their number of returned nodes to $3$. We will add this description to the final version.

**(R2) Goals of Synthetic dataset and the number of nodes $k$ in explanations.** The experiments are designed by [23]
so that we can identify which graph's components are constituted to the GNNs' predictions. Specifically, the reason for
a node's role should be a set of nodes/features in the corresponding motif. We agree with **R2** that some nodes in the
motif might not be exploited by the GNNs to compute the predictions; however, the number of returned nodes $k$ is set
to be the number of nodes in the motif because of the original experimental settings [23]. In practice, PGM-Explainer
can automatically select highly contributed nodes as the Markov-Blanket of the target variable.

**(R3) Why chose GCN [5].** We chose it because it is one of the most
well-known graph neural networks and relatively fast to train [32].

**(R3) Dependency of PGM.** Although we do not experimentally show
the accuracy and precision gains of PGM-Explainer are directly from
exploiting the dependency among features, we theoretically prove all
dependent variables must be included in the explanation (Theorem 1 and
2). Furthermore, with sufficient number of samples $n$, all dependency
statements in $\mathcal{D}_t$ must be included in the explanation graphical model.

Figure A: Predictions of GNN in experiment on Fig.1 for the central-red node with different activated features.

**(R3) Mutual information (MI) objective (line 302-303).** Since the
MI objective does not maximize the predicted label, features promoting
other classes can also maximize the objective. We demonstrate this in the same GNN considered in Fig.1 of the
manuscript. Instead of considering node $E$, we examine the prediction *red* on the central node $A$. Fig. A shows the
soft-max predictions of $A$ when different graph's components are activated/inactivated. By only activating the yellow
node (shown in Fig.1(d) in the manuscript), which is not a node in the motif, the conditional entropy of $A$'s soft-max
output reduced significantly (from the middle to the right figure of Fig. A). However, this node does not contribute
constructively to the original prediction (the second class). We observe that this phenomenon occurs more frequently
with nodes of high degree. We will smooth this expression in the final version.

**(R4) Experimental setup and Time complexity.** The models are directly taken from previous work [23] and [32]
(stated at line 250 and 267). Specifically, all models are trained using Adam optimizer. In the node classifications,
all models have 3 graph layers with the number of parameters between $1102$ and $1548$. The train/validation/test split
is $80/10/10\%$. The models are trained for $1000$ epochs with learning rate $0.001$. For the graph classification, the
dataset is divided into 55K/5K/10K. The model has $101365$ parameters and converges at epoch $188$ with learning
rate is automatically chosen between $\{10^{-3}, 10^{-4}\}$. The models' accuracy and the number of samples $n$ used by
PGM-Explainer to generate the explanations are shown in the below table. Here, $n$ is chosen based on the expected
number of variables contributing to the target prediction. If the graph maximum degree is much less than the number of
nodes, the time complexity of PGM-Explainer is dominated by $n$ forwarding computations of the model in the data
generation step. We will include this discussion in the final version.

| | **Node Classification** | | | | | | | | **Graph Classification** |
|---|---|---|---|---|---|---|---|---|---|
| **Experiment** | syn1 | syn2 | syn3 | syn4 | syn5 | syn6 | Btc-alpha | Btc-OTC | GCN-MNIST |
| **Accuracy** | 97.9 | 85.4 | 100.0 | 99.4 | 89.1 | 99.3 | 93.9 | 89.5 | 90.4 |
| **No. Samples $n$** | 800 | 800 | 800 | 1600 | 4000 | 800 | 1000 | 1000 | 400 |

Finally, we thank reviewers for constructive suggestions in including some formal definitions (**R1**: Barabasi-Albert
graph, Perfect map, Markov-Blanket...) and illustrations (**R2**: PGM with/without child constraint). We really appreciate
these suggestions and, if our work is accepted, we will include them in the final version.

[Meta-Review · NeurIPS 2020]

In this paper, the authors propose a new method to explain GNN. The proposed algorithm PGM-Explainer is technically sounds and novel. Through experiments, the authors demonstrated that the proposed method outperforms GNNexplainer, which is a state-of-the-art GNN explainer. The explanation of GNN is an important research topic and there exist a few methods. Moreover, all reviewers are positive about the paper, and thus, this paper is good to be presented at NeurIPS.